# Obstructive sleep apnea and hypopnea syndrome in patients admitted in a tertiary hospital in Cameroon: Prevalence and associated factors

**Virginie Poka-Mayap**[1]*, **Dodo Balkissou Adamou**[2], **Massongo Massongo**[1], **Steve Voufouo Sonwa**[3], **Jacqueline Alime**[3], **Ben Patrick Michel Moutlen**[3], **Alfred Kongnyu Njamnshi**[4,5,6], **Andre Noseda**[7], **Eric Walter Pefura-Yone**[1,4]

**1** Pulmonology Department, Yaounde Jamot Hospital, Yaounde, Cameroon, **2** Faculty of Medicine of Garoua, The University of Ngaoundere, Garoua, Cameroon, **3** Faculty of Medicine and Biomedical Sciences, The University of Yaounde I, Yaounde, Cameroon, **4** Department of Internal Medicine and Subspecialties, Faculty of Medicine and Biomedical Sciences, The University of Yaounde I, Yaounde, Cameroon, **5** Department of Neurology, Yaounde Central Hospital, Yaounde, Cameroon, **6** Brain Research Africa Initiative (BRAIN), Yaounde, Cameroon, **7** Pulmonology Department, Brugmann University Hospital U.L.B., Brussels, Belgium

* pokavirginie@yahoo.fr

**Data Availability Statement:** The authors confirm that all data underlying the findings are fully

## Abstract

### Purpose

Obstructive sleep apnea and hypopnea syndrome (OSAHS) is poorly documented in Sub-Saharan Africa, especially in the hospital setting. The aim of this study was to determine its prevalence and to investigate the associated factors in patients admitted in a tertiary referral hospital in Cameroon.

### Methods

In this cross-sectional study conducted in the Cardiology, Endocrinology and Neurology departments of the Yaounde Central Hospital; all patients aged 21 and older were included consecutively. A sample of randomly selected patients was recorded using a portable sleep monitoring device (PMD). OSAHS was defined as apnea-hypopnea index (AHI) $\geq$ 5/hour (with > 50% of events being obstructive) and moderate to severe OSAHS as an AHI > 15/hour. Logistic regression was used to identify factors associated to OSAHS.

### Results

Of the 359 patients included, 202 (56.3%) patients were women. The mean age (standard deviation) was 58 (16) years. The prevalence of OSAHS assessed by PMD (95% CI) was 57.7% (48.5–66.9%), 53.8% in men and 62.7% in women (p = 0.44). The median (25th-75th percentiles) AHI, body mass index and Epworth Sleepiness Scale score of OSAHS patients were 17 (10.6–26.9)/hour, 27.4 (24.7–31.6) kg/m$^2$ and 7 (5–9) respectively. The only factor

available without restriction. All relevant data are within the Supporting Information files.

**Funding:** The author(s) received no specific funding for this work.

**Competing interests:** The authors have declared that no competing interests exist.

associated to moderate to severe OSAHS was hypertension [odds ratio (95% CI)]: 3.24 (1.08–9.72), p = 0.036.

## Conclusion

OSAHS is a common condition in patients in this health care centre of Cameroon. In the hospital setting, screening for OSAHS in patients with hypertension is recommended.

## Introduction

Obstructive Sleep Apnea and Hypopnea Syndrome (OSAHS) is now recognized as a cardio-vascular risk factor responsible for significant morbidity and mortality if untreated [1–4]. Prevalence estimates depend on populations examined, time periods of assessment, and how breathing events of obstructive sleep apnea and hypopnea are defined, but can reach large minorities to majorities of specific population subgroups[5].

The prevalence of OSAHS defined as an apnea hypopnea index (AHI) $\geq$ 5 per hour has varied from 6% in men and 4% in women (1993) to 83.8% in men and 60.8% in women (2013)[6–8]. In developing countries, particularly in sub-Saharan Africa (SSA), the prevalence of high risk of OSAHS, assessed by questionnaires, is between 30 and 60%[9–11]. Using portable sleep monitoring devices (PMD) or polysomnography, the prevalence of the disease in the general population is 28.5% and that of moderate to severe OSAHS is 6.3% while the prevalence is between 40–60% in patients attending hospital [10,12,13].

The prevalence of OSAHS is even larger in health care setting. It amounts 40 to 80% in developed countries [14–17]. In SSA, few studies document the prevalence of OSAHS in health care setting [9,18]. In daily clinical practice, patients with high risk of OSAHS are frequently admitted to hospital, but the OSAHS is rarely detected and, as a consequence, subsequent management is not optimal [19]. Knowing the prevalence of OSAHS in our hospital patients will enable to take appropriate measures to improve the diagnosis and the overall management of patients. The aim of this study was to evaluate the prevalence of OSAHS and to investigate associated factors in patients admitted in a tertiary referral health care centre in Cameroon.

## Materials and methods

### Participants and study setting

This cross-sectional study, conducted over a 7-month period from November 2016 to May 2017, was carried out at the Yaounde Central Hospital (YCH), a government-run tertiary health facility in the capital city of Cameroon, which serves as referral centre for Yaounde and its environs. All patients aged 21 and older admitted in the Cardiology, Neurology and Endocrinology departments of the YCH were eligible for the study. Patients unable to answer questionnaires, with nasogastric tubes were excluded. The participants were recruited consecutively during the inclusion period.

### Procedure

All eligible patients who signed an informed consent completed a STOP-BANG questionnaire. They were classified into two groups according to the risk of OSAHS (low versus high) as assessed from the STOP-BANG questionnaire. Every day, a participant was randomly selected from each group to be recorded by a portable sleep monitoring device (PMD).

## Data collection

The socio-demographic characteristics recorded were: age, sex, marital status and occupational category. The comorbidities recorded were hypertension, diabetes mellitus, heart failure, stroke, human immunodeficiency virus (HIV) infection, and epilepsy. Alcohol consumption was assessed as follows: current regular, occasional, former consumer (at least 12 months without alcohol consumption) or never consumer of alcohol and the ethylic index was calculated in grams per day [amount of alcohol consumed daily in ml x alcohol percentage x density (0.8)]. Tobacco smoking was assessed and participants ranked as current smokers (participants having smoked at least one cigarette a day for at least one year, or having smoked at least 20 cigarettes in their lifetime and still smoking), ex-smokers (participants who declared having stopped smoking for at least six months) and never-smokers.

The following symptoms were collected were: snoring, apnea witnessed by family members, nocturia, daytime fatigue, daytime sleepiness, morning headaches, non-restorative sleep. The daytime sleep propensity was assessed by the Epworth Sleepiness Scale (ESS) which consists of eight items (described situations) arranged on a 4-point Likert scale ranging from 0 ('never doze') to 3 ('high chance of dozing' during daytime). The summed scores range from 0 to 24; scores above 10 are commonly interpreted as consistent with excessive daytime sleepiness [20]. Neck circumference, hip circumference, weight, height were measured and body mass index (BMI) derived as the ratio: weight (kg) / height / height (m$^2$). A BMI $\geq 30$ kg / m$^2$ defines obesity.

## OSAHS assessment

**Screening for risk of OSAHS.** The risk of OSAHS was evaluated by the STOP-BANG questionnaire, a tool validated in different clinical situations for OSAHS screening [21]. It consists of 8 questions that can be answered by "yes" or "no". The number of "yes" answers determines the questionnaire score. The risk of OSAHS is high when the score $\geq 3$ [22].

**Diagnosis of OSAHS.** Respiratory events during sleep were recorded by PMD (SLEEP FAIRY Lt, Co), a type III recording system. The PMD was placed on the participant in hospital at bedtime and the recording was started by the participant or his/her family member at the time of falling asleep and stopped at awakening. The signals recorded included airflow through a nasal pressure sensor, oxyhaemoglobin saturation and pulse rate from pulse oximetry as well as thoracic and abdominal movements through respiratory inductance plethysmography. The sleep recordings were manually scored by a certified polysomnologist, with appropriate sleep related training. Recordings with unreadable signal on more than 25% of the recording time and recordings of less than 4 hours were excluded. The respiratory events were scored according to the "The AASM Manual for the Scoring of Sleep and Associated Events"[23]. Obstructive sleep apnea was defined as a reduction $\geq 90\%$ in airflow for $\geq 10$ seconds during sleep with persistent ventilatory efforts. Hypopnea was defined as a reduction of $\geq 30\%$ airflow for $\geq 10$ seconds during sleep associated with a desaturation $\geq 3\%$. The apnea-hypopnea index (AHI) was the result of the ratio of the sum of all apneas and hypopneas divided by the total recording time. According to the "International Classification of Sleep Diseases" 3rd version criteria, OSAHS was diagnosed when the AHI was $\geq 5$ per hour of recording in presence of OSAHS symptoms with $> 50\%$ of events being obstructive, and moderate to severe OSAHS when the AHI was $\geq 15$ per hour[24].

## Ethical statement

The study was approved by the Ethics Committee of the Faculty of Medicine and Biomedical Sciences of the University of Yaounde I, Cameroon. The study received administrative

authorisation from the Centre Region Delegation of the Ministry of Public Health of Cameroon. Subsequently, a recruitment authorization from the YCH administrative team was obtained, as well as the written informed consent of all included participants.

## Statistical analysis

The sample size was determined assuming that two participants a day would be recorded by PMD during the study period, thus 400 subjects. Assuming a 40% prevalence of OSAHS in in-hospital patients [17], a type 1 error of 5%, a power of 80%, the required sample size was 182 participants.

Data were analysed using IBM-SPSS Version 20 for Windows (SPSS Inc., Chicago, IL). Qualitative data are presented as counts and proportions, and quantitative variables as mean and standard deviation (SD) or median and $25^{th}$-$75^{th}$ percentiles. Chi squared test and Fisher exact test were used to compare proportions. Quantitative variables were compared by Student's T test or its non-parametric equivalent. The prevalence of OSAHS was that obtained in the sub-sample of patients who were recorded by PMD. Logistic regression models were used to investigate factors associatedto moderate to severe OSAHS. Potential associated factors were first tested in univariate analysis, and significant ones (based on $p < 0.10$ threshold) were further tested in multivariate models. A p-value $< 0.05$ was used to define statistically significant results.

## Results

As shown in Fig 1, a total of 383 subjects were invited to take part in the study, of whom 24 declined (response rate of 93.7%). Three hundred and fifty-nine subjects participated in the

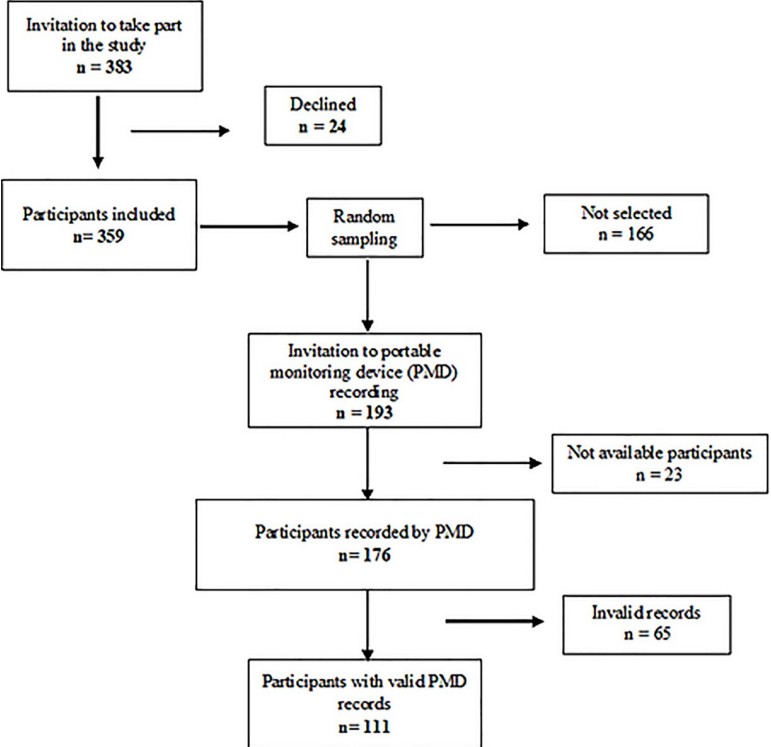

**Fig 1. Flow chart of the inclusion of participants.** PMD: Portable Monitoring Device.

study, from which 193 patients were randomly selected for PMD recording. Only 176 subjects were recorded, of which data were valid in 111 subjects.

## General characteristics of the study population

Of the 359 participants, 202 (56.3%) were women. The mean age (standard deviation) was 58 (16) years. As presented in Table 1, the most common comorbidities were hypertension (50.1%) and diabetes mellitus (34.3%). Hypertension was significantly more frequent in women than men. Current smoking and regular or occasional alcohol consumption were found in 8.6% and 58.2% of participants respectively.

The Table 2 summarises the clinical characteristics of participants. Snoring was the most common symptom of OSAHS and was significantly more prevalent in men. Two hundred and five (57.1%) participants reported sleepiness, while only 24 (6.7%) had an ESS consistent with excessive daytime sleepiness. Obesity was present in 30.6% of patients and more prevalent in women than men (28% vs. 21%, p = 0.001). The neck circumference of men was higher than that of women (39 vs. 35 cm, p < 0.001).

## Prevalence of high risk of OSAHS

As shown in Fig 2, The prevalence [95% confidence interval (95% CI)] of high risk of OSAHS, as assessed from the STOP-BANG questionnaire, was 64.1% (59.1–69.1%) in the whole population, 78.3% (71.9–81.7%) in men and 53% (46.1–59.9%) in women (p < 0.001).

**Table 1. Sociodemographic characteristics and comorbidities of the study population.**

| Characteristics | Overall n = 359 (%) | Men n = 157 (%) | Women n = 202 (%) | p-value |
|---|---|---|---|---|
| **Age in years, mean (SD)** | 58 (16) | 56 (15) | 59 (16) | 0.059 |
| **Marital status** | | | | |
| Alone | 173 (48.2) | 39 (24.8) | 134 (66.3) | <0.001 |
| Living in couple | 186 (51.8) | 118 (75.2) | 68 (33.7) | |
| **Occupational category** | | | | |
| Unemployed | 321 (89.4) | 129 (17.8) | 192 (96) | <0.001 |
| Employed | 38 (10.6) | 28 (82.2) | 10 (4) | |
| **Hypertension** | 180 (50.1) | 67 (42.7) | 113 (55.9) | 0.014 |
| **Diabetes mellitus** | 123 (34.3) | 55 (35) | 68 (33.7) | 0.823 |
| **Heart failure** | 53 (14.8) | 20 (12.7) | 33 (16.3) | 0.371 |
| **Stroke** | 111 (31) | 55 (35.3) | 56 (15.6) | 0.135 |
| **HIV infection** | 48 (13.4) | 15 (9.6) | 33 (16.3) | 0.085 |
| **Epilepsy** | 10 (2.8) | 7 (4.5) | 3 (1.5) | 0.111 |
| **Tobacco smoking** | | | | |
| Smokers | 31 (8.6) | 17 (10.8) | 14 (6.9) | <0.001 |
| Ex-smokers | 30 (8.4) | 24 (15.3) | 6 (3) | |
| Non-smokers | 298 (83) | 116 (73.9) | 182 (90.1) | |
| **Alcohol consumption** | | | | |
| Regular | 65 (18.1) | 39 (24.8) | 26 (12.9) | <0.001 |
| Occasional | 144 (40.1) | 78 (49.7) | 66 (32.7) | |
| Old consumer | 78 (21.7) | 26 (16.6) | 52 (25.7) | |
| Never | 72 (20.1) | 14 (8.9) | 58 (28.7) | |
| **Ethylic index, g/day, median in (25th-75th percentiles)** | 16.5 (2.9–57.2) | 28.6 (5.7–93.8) | 2.86 (0.2–28.6) | <0.001 |

SD: Standard Deviation; HIV: Human Immunodeficiency Virus.

**Table 2. Clinical characteristics of the study population.**

| Characteristics | Overall n = 359 (%) | Men n = 157(%) | Women n = 202(%) | p-value |
|---|---|---|---|---|
| **OSAHS symptoms** | | | | |
| Snoring | 215 (59.9) | 109 (69.4) | 106 (52.5) | 0.001 |
| Apnea witnessed | 64 (17.8) | 37 (23.6) | 27 (13.4) | 0.018 |
| Nocturia | 141 (39.3) | 94 (59.9) | 124 (61.4) | 0.828 |
| Daytime fatigue | 293 (81.6) | 128 (81.5) | 165 (81.7) | 1 |
| Non-restorative sleep | 44 (31.8) | 53 (33) | 31 (31) | 0.494 |
| Sleepiness | 205 (57.1) | 91 (58) | 114 (56.4) | 0.83 |
| Morning headaches | 140 (39) | 54 (34.4) | 86 (42.6) | 0.127 |
| **BMI, kg/m$^2$** | 26.1 (22.3–30.8) | 25.7 (23.2–29.3) | 26.4 (21.3–32.9) | 0,31 |
| **BMI > 30 kg/m$^2$** | 110 (30.6) | 33 (21) | 74 (28.1) | 0.001 |
| **Hip circumference, cm** | 90 (78–102) | 88 (80–99) | 94 (76–108) | 0.047 |
| **Neck circumference, cm** | 37 (34–39) | 39 (37–41) | 35 (33–37.5) | <0.001 |
| **ESS** | 6 (3–8) | 7 (5–9) | 5 (3–7) | <0.001 |
| **ESS score > 10** | 24 (6.7) | 12 (7.6) | 12 (5.9) | 0.531 |

All continuous variables are expressed as median (25th-75th percentiles); OSAHS: Obstructive Sleep Apnea and Hypopnea Syndrome; BMI: Body Mass Index; ESS: Epworth Sleepiness Scale

## Prevalence and characteristics of OSAHS

One hundred and eleven (30.9%) participants with valid PMD records constituted the sub-sample in which the prevalence of OSAHS was objectively evaluated. In those participants, hypertension, diabetes mellitus, snoring, non-restorative sleep and high risk of OSAHS were more frequent (Table 3).

The prevalence of OSAHS (95% CI) was 57.7% (48.5–66.9%). As presented in Table 4, of the 65 patients diagnosed with OSAHS, 35 had moderate to severe OSAHS, giving a prevalence (95% CI) of 31.5% (22.9–40.1%). OSAHS was found in 37 (62.7%) women vs 28 (53.8%) men (p = 0.44) and the equivalent figure for moderate to severe OSAHS was 22 (37.3%) women vs 13 (25%) men (p = 0.22). The median (25th-75th percentiles) AHI, BMI and ESS score of OSAHS patients were respectively 17 (10.6–26.9) per hour, 27.4 (24.7–31.6) kg/m$^2$ and 7 (5–9). The median arterial oxygen saturation was lower in women and women had a lower ESS score. None of the participants diagnosed with OSAHS was previously aware of this diagnosis.

## Factors associated to moderate to severe OSAHS

Patients with moderate to severe OSAHS (n = 35) were compared to those without sleep apnea syndrome (n = 40) as presented in Table 5. In univariate analysis, hypertension was present in 82.9% of patients with moderate to severe OSAHS and in 52.9% of patients unaffected (p = 0.004). Forty percent of patients with moderate to severe OSAHS were living alone compared to 20% of those without OSAHS (p = 0.032). In multivariate analysis, the only factor associated with moderate to severe OSAHS [odds ratio (95% CI)] was hypertension [3.24 (1.08–9.72)]. Exclusion of OSAHS symptoms as potential associated factors did change neither the direction neither the strength of the association between moderate to severe OSAHS and its potentially associated factors.

## Discussion

The aim of this cross-sectional study was to determine the prevalence of OSAHS and to investigate factors associated to OSAHS in patients admitted in a tertiary hospital in Cameroon.

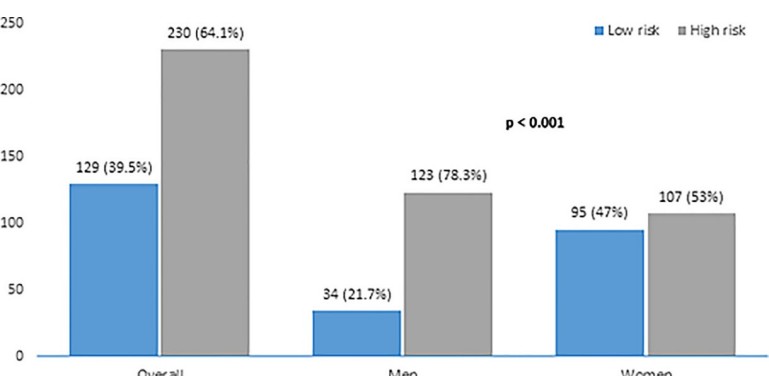

**Fig 2. Risk of obstructive sleep apnea and hypopnea syndrome assessed by STOP-BANG questionnaire in the population study.**

The main results are: 1) a high risk of OSAHS was present in 64% of participants, as assessed from the STOP-BANG questionnaire; 2) about 2/3 of participants were diagnosed with OSAHS, while 1/3 had moderate to severe OSAHS; 3) hypertension was the only factor associated to moderate to severe OSAHS.

The prevalence of high risk of OSAHS we found in the present study using the STOP-BANG questionnaire is as high as previously reported by Sharma et al in 2015, in a study including obese patients admitted in a department of internal medicine [14]. In another study conducted by Njamnshi et al in Cameroon, 60% of patients with hypertension treated as outpatients were found to have a high risk of OSAHS using the Berlin questionnaire [25]. In the present study including patients with several comorbidities, the risk of OSAHS was found to be much higher than that assessed in the general population using the same questionnaire; namely 17.8% in Cameroon [10] or 36.9% in Nigeria [9].

**Table 3. Characteristics of participants recorded and not recorded by the portable sleep monitoring device.**

| Characteristics | Recorded by PMD n = 111 (%) | Not recorded by PMD n = 248 (%) | p-value |
|---|---|---|---|
| Women | 59 (53.2) | 143 (57.7) | 0.49 |
| Age in years, mean (SD) | 58 (12) | 57 (17) | 0.69 |
| Hypertension | 70 (63.1) | 110 (44.1) | 0.001 |
| Diabetes mellitus | 49 (44.1) | 74 (29.8) | 0.011 |
| Heart failure | 20 (18) | 33 (13,3) | 0.371 |
| Stroke | 38 (34.5) | 73 (29.4) | 0.262 |
| Snoring | 79 (71.2) | 136 (54.8) | 0.004 |
| Apnea witnessed | 27 (24.3) | 37 (14.9) | 0.037 |
| Daytime fatigue | 95 (85.6) | 198 (79.8) | 0.238 |
| Non-restorative sleep | 44 (39.6) | 70 (28.9) | 0.037 |
| Sleepiness | 71 (64) | 134 (54) | 0.083 |
| Morning headaches | 47 (42.3) | 93 (37.5) | 0.413 |
| BMI > 30 kg/m$^2$ | 42 (37.8) | 68 (27.4) | 0.063 |
| Neck circumference, cm (25th-75th percentiles) | 37.5 (34–41) | 36 (34–39) | 0.106 |
| Daytime sleepiness (ESS >10) | 10 (9) | 14 (5.6) | 0.257 |
| STOP-BANG $\geq$ 3 | 85 (76.6) | 145 (58.5) | 0.001 |

SD: Standard Deviation; BMI: Body Mass Index; PMD: Portable sleep Monitoring Device; OSAHS: Obstructive Sleep Apnea and Hypopnea Syndrome; ESS: Epworth Sleepiness Scale

**Table 4. Characteristics of patients diagnosed with obstructive sleep apnoea and hypopnea syndrome.**

| Characteristics | Overall n = 65 (%) | Men n = 28 (%) | Women n = 37 (%) | p-value |
|---|---|---|---|---|
| Total recording time, min | 420 (360–480) | 420 (360–450) | 480 (360–480) | 0.026 |
| Number of OA | 38 (12–73) | 28 (11–69) | 48 (15–114) | 0.25 |
| Number of hypopnea | 55 (32–92) | 58 (29–92) | 54 (36–92) | 0.88 |
| Mean SaO2, % | 94 (92–96) | 95 (93–96) | 94 (92–95) | 0.043 |
| Minimal SaO2, % | 81 (73–86) | 81 (74–88) | 82 (72–85) | 0.67 |
| ODI, $h^{-1}$ | 23 (15–36) | 23 (13–38) | 23 (16–35) | 0.62 |
| AHI, $h^{-1}$ | 17 (10.6–26.9) | 14.1 (10.8–26.6) | 17.9 (9.5–29.6) | 0.89 |
| AHI > 15 | 35 (53.8) | 13 (46.4) | 22 (59.4) | 0.33 |
| Ethylic index, g/day | 28.6 (2.86–59.8) | 57.2 (6.2–120.3) | 2.9 (0–28.6) | 0.015 |
| Sleepiness | 47 (73.3) | 21 (75) | 26 (70.3) | 0.79 |
| BMI, $kg/m^2$ | 27.4 (24.7–31.6) | 27.6 (24.7–30.9) | 27.3 (24–31.6) | 0.88 |
| BMI > 30 $kg/m^2$ | 40 (61.5) | 20 (71.4) | 20 (54.1) | 0.20 |
| ESS score | 7 (5–9) | 8 (5–10) | 6 (4–9) | <0.001 |
| ESS score > 10 | 9 (13.8) | 4 (14.3) | 5 (13.5) | 1 |

All continuous variables are expressed as median (25$^{th}$-75$^{th}$ percentiles); OSAHS: Obstructive Sleep Apnea and Hypopnea Syndrome; AHI: Apnea-hypopnea index; SaO2: arterial oxygen saturation; ODI: oxygen desaturation index; OA: Obstructive apnea; ESS: Epworth Sleepiness Scale; BMI: Body Mass Index

To the best of our knowledge, few studies until now have characterised OSAHS in in-hospital patients in SSA. Other studies using PMD as diagnostic tool for OSAHS in hospitalized patients found that the prevalence of OSAHS was between 40 and 80% [16–18]. The prevalence of OSAHS in this cross-sectional study was 57.7%. An even higher prevalence of 87%

**Table 5. Factors associated to moderate to severe OSAHS.**

| Factors | n (%) | Univariate analysis | | Multivariate analysis | |
|---|---|---|---|---|---|
| | | OR (95% CI) | p-value | OR (95% CI) | p-value |
| Male sex | 13 (37.1) | 0.62 (0.27–1.43) | 0.268 | - | - |
| Age > 50 years | 31 (88.6) | 1.58 (1.35–1.73) | 0.05 | 1.81 (0.50–6.58) | 0.37 |
| Living alone | 14 (40) | 2.66 (1.10–6.52) | 0.032 | 1.50 (0.46–4.93) | 0.51 |
| Illeterate | 31 (88.6) | 1.67 (0.42–6.68) | 0.463 | - | - |
| Employed | 17 (48.6) | 1.19 (0.53–2.68) | 0.678 | - | - |
| Ethylic index > 30 g/day | 14 (40) | 0.6 (0.26–1,41) | 0.24 | - | - |
| Hypertension | 29 (82.9) | 4.31 (1.59–11.68) | 0.004 | 3.24 (1.08–9.72) | 0.036 |
| Heart failure | 10 (28.6) | 2.7 (0.98–7.47) | 0.054 | 1.94 (0.63–6.01) | 0.25 |
| Stroke | 10 (28.6) | 0.79 (0.33–1.94) | 0.619 | - | - |
| Snoring | 27 (77.1) | 1.65 (0.65–4.20) | 0.292 | - | - |
| Apnea witnessed | 9 (24.7) | 1.17 (0.45–2.99) | 0.746 | - | - |
| Daytime fatigue | 33 (94.3) | 3.76 (0.79–17.71) | 0.094 | 2.53 (0.48–13.22) | 0.27 |
| Non-restorative sleep | 18 (51.4) | 2.03 (0.89–4.64) | 0.093 | 2.05 (0.82–5.13) | 0,13 |
| Sleepiness | 26 (74.3) | 2.04 (0.83–5) | 0.118 | - | - |
| Morning headaches | 19 (54.3) | 1.89 (0.83–4.30) | 0.128 | - | - |
| ESS score > 10 | 6 (17.1) | 3.41 (0.89–13.01) | 0.072 | 1.75 (0.41–7.53) | 0.45 |
| Neck circumference ≥ 38 cm | 18 (51.4) | 0.75 (0.33–1.69) | 0.49 | - | - |
| BMI > 30 $kg/m^2$ | 24 (68.4) | 1.54 (0.65–3.62) | 0.321 | - | - |

95% CI: 95% Confidence Interval; OR: Odds Ratio; OSAHS: Obstructive Sleep Apnoea and Hypopnoea Syndrome; ESS: Epworth Sleepiness Scale

was found in obese patients at high risk of OSAHS at Philadelphia, using polysomnography, the gold standard for the diagnosis of OSAHS [14,26]. These prevalence rates are higher than those found in the community, probably because of the existence of an association between OSAHS and conditions leading to hospitalization, as already reported in other studies[27,28]. Furthermore, 31.5% of the participants had moderate to severe OSAHS, potentially requiring specific management. However, none of these patients were aware of their diagnosis of OSAHS, corroborating the results of previous works that underline the under-diagnosis of OSAHS in health care centres [14,29].

Factors associated to OSAHS in a hospital setting may differ according to the comorbidities present in the study population. In a Chinese study including type 2 diabetes patients, a per 1 kg/m$^2$ increase in BMI and a per 1 year increase in age were associated to OSAHS [16]. The ESS score $\geq$ 4 and nocturnal desaturation < 82% in hospitalized patients for acute myocardial infarction in Tunisia were associated with OSAHS [18]. In our study hypertension was independently associated to moderate to severe OSAHS, with an odd ratio as high as 3.24 in a sample where 63.1%, 44.1%, 34.5% and 18% respectively presented with hypertension, diabetes mellitus, stroke and heart failure. In a group of patients hospitalized for the management of a metabolic syndrome with controlled hypertension in France, Hansel et al also found hypertension to be associated to moderate to severe OSAHS [17].

The main limitation of the present study is that only one-third of the participants initially interviewed were recorded by PMD. Despite the random selection of patients to be recorded, while taking in consideration the risk of OSAHS, the two groups (recorded vs. non-recorded) were not entirely similar. Indeed, the frequency of known comorbidities of OSAHS (hypertension and diabetes mellitus), symptoms of OSAHS (snoring, non-restorative sleep) and high risk of OSAHS were greater in the recorded patients, probably justifying the high prevalence of OSAHS. Another limitation is the use of PMD as a diagnostic tool for OSAHS, and not the gold standard which is polysomnography. In our country, only one centre has a polysomnograph. Polysomnography, which requires more resources, is thus not very accessible. Although using PMD for the type of study that we performed seems to be appropriate, further research using the gold standard is necessary in the future. Nevertheless, PMD has a good sensitivity in high risk patients and / or with a strong clinical presumption of OSAHS [26], which was the case in this work. Furthermore, PMD yields a lower than expected AHI, since the total recording time is higher than the total sleep time and as PMD does not identify micrarousals, another diagnostic criteria of the hypopneas [26]. All this suggests that the prevalences obtained in this study may be plausible. To our knowledge, our study is the first in Cameroon, and even in whole SSA, to assess OSAHS through respiratory recordings during sleep and to evaluate the factors associated to OSAHS in a large group of hospitalized patients.

## Conclusion

OSAHS is a common and under-diagnosed condition in patients with comorbidities in this Cameroonian tertiary health care centre. Hypertension is a factor significantly associated to moderate to severe OSAHS. In hospital setting, the screening for OSAHS in patients with hypertension, particularly if resistant or uncontrolled, is recommended. Additional studies, preferably using polysomnography for the diagnosis of OSAHS, are needed to confirm and extend the results obtained in this study.

## Supporting information

**S1 Data. Database of all patients included in the study.**
(XLSX)

**S2 Data. Database of randomly selected patients, recorded by portable sleep monitoring device in the study.**
(XLSX)

## Acknowledgments

The work was performed with the support of Nadine Kanko Nguekam.

## Author Contributions

**Conceptualization:** Virginie Poka-Mayap, Dodo Balkissou Adamou, Massongo Massongo, Eric Walter Pefura-Yone.

**Data curation:** Virginie Poka-Mayap.

**Formal analysis:** Virginie Poka-Mayap.

**Investigation:** Steve Voufouo Sonwa, Jacqueline Alime, Ben Patrick Michel Moutlen.

**Methodology:** Dodo Balkissou Adamou.

**Supervision:** Dodo Balkissou Adamou, Massongo Massongo, Alfred Kongnyu Njamnshi, Eric Walter Pefura-Yone.

**Writing – original draft:** Virginie Poka-Mayap.

**Writing – review & editing:** Alfred Kongnyu Njamnshi, Andre Noseda, Eric Walter Pefura-Yone.

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
