## [Decision Letter · Decision Letter 0]

9 Oct 2019

PONE-D-19-25768

Obstructive sleep apnea and hypopnea syndrome in patients admitted in a tertiary hospital in Cameroon: prevalence and determinants

PLOS ONE

Dear Dr. Poka-Mayap,

Thank you for submitting your manuscript to PLOS ONE. After careful consideration, we feel that it has merit but does not fully meet PLOS ONE’s publication criteria as it currently stands. Therefore, we invite you to submit a revised version of the manuscript that addresses the points raised during the review process.

Please take a close look at the comments of Reviewer #1 who has made many specific comments that will improve the readability of the manuscript.  

In addition to the above, this editor has questions re: the analysis for moderate-severe OSA.  It is not clear to me why BMI is not included in the analysis and why an age>58 was chosen as a cutoff. Also, the authors could consider using the STOP-BANG as a factor for prediction as the STOP-BANG has been shown to be predictive of moderate-severe OSA.

We would appreciate receiving your revised manuscript by Nov 23 2019 11:59PM. To enhance the reproducibility of your results, we recommend that if applicable you deposit your laboratory protocols in protocols.io, where a protocol can be assigned its own identifier (DOI) such that it can be cited independently in the future. For instructions see: http://journals.plos.org/plosone/s/submission-guidelines#loc-laboratory-protocols

We look forward to receiving your revised manuscript.

Kind regards,

James Andrew Rowley

Academic Editor

PLOS ONE

Journal Requirements:

1. Please include additional information regarding the STOP-BANG questionnaire used in the study and ensure that you have provided sufficient details that others could replicate the analyses. For instance, if you developed a questionnaire as part of this study and it is not under a copyright more restrictive than CC-BY, please include a copy, in both the original language and English, as Supporting Information.

2. In your Methods section, please provide additional information about the participant recruitment method and the demographic details of your participants. Please ensure you have provided sufficient details to replicate the analyses such as: a) the recruitment date range (month and year), b) a description of any inclusion/exclusion criteria that were applied to participant recruitment,  and c) a description of how participants were recruited.

Reviewers' comments:

Reviewer's Responses to Questions

**Comments to the Author**

1. Is the manuscript technically sound, and do the data support the conclusions?

Reviewer #1: Yes

Reviewer #2: Yes

2. Has the statistical analysis been performed appropriately and rigorously? 

Reviewer #1: Yes

Reviewer #2: Yes

3. Have the authors made all data underlying the findings in their manuscript fully available?

Reviewer #1: No

Reviewer #2: No

4. Is the manuscript presented in an intelligible fashion and written in standard English?

Reviewer #1: Yes

Reviewer #2: Yes

5. Review Comments to the Author

Reviewer #1: 1.AHI should replace OAHI : the standard abbreviation is AHI (Apnea hypopnea index) which makes your article more searchable. Also unless you have excluded the few central events which may coexist with obstructive events in OSAHS, then OAHI is a misnomer.

2. Rephrase last sentence in the conclusion of the abstract " Screening...in patients especially "in those with" hypertension....

3. Abstract conclusion: I do not agree screening is mandatory in all in patients with hypertension but rather in 1) those with resistant hypertension and 2) in those with hypertension and additional risk factors for .

4. Line 66 and 68 of introduction: Define "sleep records", describe what you mean by "particular groups of patients" so the readers understand exactly what you are referring to.

5. Line 105 in data collection: propensy should be "propensity"; after ESS insert "which"

6. Line 126 of Diagnosis of OSAHS: what is illisible? likely a typo, please correct

7. Line 131 of Diagnosis of OSAHS: consider taking out obstructive and using the universal term "Apnea hypopnea index" except all central events were excluded from the calculation

8. Line 132 of Diagnosis of OSAHS: consider changing total registration time to "total recording time" which is a more universal term. Thiswill need to be changed throughout the manuscript and in the tables if applicable.

9. Table 1: please correct all p values with commas inserted instead of decimal points. These are probably erroneous. Also ensure p values are placed in the appropriate rows for which they were calculated.

10. Table 2: again correct p values with commas instead of decimal points

11. Line 227: please rephrase, using the word "but" suggests that a lower median arterial oxygen saturation is associated with lower ESS scores, which it is not. Consider expressing these 2 findings as separate sentences.

12. Table 4: if changing total registration time to total recording time as discussed above, then correct it in this table, also OAHI to AHI. also correct p values with commas instead of decimal points. Change this in Lines 233 and 234 as well (OAHI to AHI)

13. Lines 238 to 239: The numbers do not add up. If 111 had valid data per the Results on line 162 then 6 particants are unaccounted for (40 - No OSA, 65 had OSAHS (30 had mild OSA, 35 with moderate to severe OSA)). Was there any other missing data? I missed any statements on further exclusions after the total of 111.

14. Lines 244-246. Please explain or rephrase. This sentence is confusing and has no clear meaning as it is written.

15. Table 5. why age > 58 years? How did you come about 58?? was there something in the data to suggest that 58 was an age that impacted having OSAHS or was it just based on the calculated mean age?

I believe age > 50 is what standard questionnaires use. Or if you want to look at teh geriatric population vs the other adults then maybe age of 65 and above etc. 58 appears random. Also, may be better to use "BMI>30" in the last row of the table rather than the term obesity.

16. Line 287: consider replacing "underline" with "depict"

17. Line 288 - 294: Starting from the sentence "Other studies..."These lines should follow line 284 for the sake of continuity.

18. Lines 285 - 288 ending with ref (14,26). should come after line 294. You may choose to say " Further more in our study....." to bring the readers attention back to the fact that you are referring to the findings from your study.

19. Line 303: Remove "a" before controlled hypertension, did you mean uncontrolled or controlled hypertension?

20. Line 313: The polysomnogram is not time consuming (takes the same amount of time as a portable study - in both cases overnight). More appropriate to say it is more expensive and requires more resources (skilled staff, sleep lab, equipment etc)

21. Line 317: Please replaced with AHI. and total recording time

22. Consider adding to discussion RE: results lines 179 - 180 state that ESS scores did not correlate with sleepiness. This is corroborated by several papers. Examples to look at include: 1. Smith et al . Multiple dimensions of excessive daytime sleepiness J Thorac Dis 2018, 2. Silva et al. identifiction of patients with sleep disordered breathing: comparing the four-variable screening tool... J Clin Sleep med 2011. etc. It may be nice to mention that your study finding corroborates the fact that the ESS should not be used in isolation to determine the presence of sleepiness.

23. Line 327 - 328 in conclusion: Amend to say "in patients with resistant/uncontrolled hypertension in hospital settings" . It is not cost effective or required in all hospitalized patients with hypertension. Patients with hypertension and risk factors for OSAHS can be studied in the outpatient setting.

Final comments: Overall a very good paper with excellent potential to publish regarding the OSA experience in Subsaharan Africa.

Reviewer #2: Interesting study, I commend the authors to undergo such an investigation in a country in which sleep services are not as developed as in other parts of the world.

Several comments and recommendations are listed here;

1. Have the ESS and STOP-BANG questionnaires been translated and validated in local languages? If so, how does one explain the major discrepancy between the prevalence of sleepiness (assessed qualitatively) and that of ESS>10 (Table 4)?

2. In the study flow diagram - you list 65 invalid PMD records; this is a high rate - one could want to see those repeated, so that power is not affected by the significantly lower sample size analyzed.

3. Abstract (results section) and manuscript: I would avoid causal inferences such as hypertension being the determinant of moderate to severe OSA... It may not be correct and you may not be able to prove it in a cross-sectional study. It is an association.

4. Abstract (conclusion) and manuscript: too strong a language ('is mandatory...') as a conclusion to this study's findings.

6. PLOS authors have the option to publish the peer review history of their article (what does this mean?). If published, this will include your full peer review and any attached files.

Reviewer #1: No

Reviewer #2: No

---

## [Author Response · Author response to Decision Letter 0]

20 Nov 2019

Response to reviewers

To

The Editorial office

PLOS ONE

Submission of a revised manuscript

Manuscript ID: PONE-D-19-25768 entitled " Obstructive sleep apnea and hypopnea syndrome in patients admitted in a tertiary hospital in Cameroon: prevalence and associated factors"

By Virginie Poka-Mayap et al

Dear Editor,

We are grateful to the managing editor and reviewers for their time and comments on our manuscript (referenced above). We have addressed their comments and queries and would like to submit an updated version of our manuscript for publication in the Journal. Changes in the main document have been inserted with the use of red and blue color. In addition we are providing below a point-by-point response to each query from the reviewers. 

We look forward to the outcome of our submission.

Journal requirements

1. Please include additional information regarding the STOP-BANG questionnaire used in the study and ensure that you have provided sufficient details that others could replicate the analyses. For instance, if you developed a questionnaire as part of this study and it is not under a copyright more restrictive than CC-BY, please include a copy, in both the original language and English, as Supporting Information

Our answer: Thank you for raising this point. We have not developed STOP-BANG questionnaire. As mentioned in the manuscript, this screening questionnaire for OSAHS have been validated (see in the reference list: number 21). 

2. In your Methods section, please provide additional information about the participant recruitment method and the demographic details of your participants. Please ensure you have provided sufficient details to replicate the analyses such as: a) the recruitment date range (month and year), b) a description of any inclusion/exclusion criteria that were applied to participant recruitment, and c) a description of how participants were recruited.

Our answer: This has been fixed. The recruitment date range was already mentioned in our initial submission and exclusion criteria were have been added. We can read now on line 85: “Patients unable to answer questionnaires, with nasogastric tubes were excluded” 

From the Reviewer 1

1. AHI should replace OAHI : the standard abbreviation is AHI (Apnea hypopnea index) which makes your article more searchable. Also unless you have excluded the few central events which may coexist with obstructive events in OSAHS, then OAHI is a misnomer.

Our answer: This has been fixed. 

2. Rephrase last sentence in the conclusion of the abstract " Screening...in patients especially "in those with" hypertension...

Our answer: This has been fixed. In the conclusion we can now read: “Screening for OSAHS in patients with hypertension is recommended to improve the overall management of these patients”

3. Abstract conclusion: I do not agree screening is mandatory in all in patients with hypertension but rather in 1) those with resistant hypertension and 2) in those with hypertension and additional risk factors for.

Our answer: Thank you for raising this point. In this work on patients admitted to hospital, that had cardiovascular risk factors, we found that hypertension was independently associated to moderate to severe OSAHS, with an odd ratio as high as 3.24. That is why we conclude that in the context of patients admitted to hospital, we need to particularly screen for OSAHS in this group of patients. .

4. Line 66 and 68 of introduction: Define "sleep records", describe what you mean by "particular groups of patients" so the readers understand exactly what you are referring to.

Our answer: Thank for the remark. We have expanded the section which now reads: “Using portable monitoring device (PMD) or polysomnography, the prevalence of the disease in the general population is 28.5%, and that of moderate to severe OSAHS is 6.3%, while the prevalence is between 40-60% in patients attending hospital [10,12,13]”

5. Line 105 in data collection: propensy should be "propensity"; after ESS insert "which"

Our answer: This has been fixed.

6. Line 126 of Diagnosis of OSAHS: what is illisible? likely a typo, please correct

Our answer: This has been fixed. We can now read: “Recordings with unreadable signal on more than 25% of the recording time and recordings of less than 4 hours were excluded”

7. Line 131 of Diagnosis of OSAHS: consider taking out obstructive and using the universal term "Apnea hypopnea index" except all central events were excluded from the calculation

Our answer: This has been fixed.

8. Line 132 of Diagnosis of OSAHS: consider changing total registration time to "total recording time" which is a more universal term. This will need to be changed throughout the manuscript and in the tables if applicable.

Our answer: This has been fixed.

9. Table 1: please correct all p values with commas inserted instead of decimal points. These are probably erroneous. Also ensure p values are placed in the appropriate rows for which they were calculated.

Our answer: This has been fixed. All commas have been replaced by decimal points.

10. Table 2: again correct p values with commas instead of decimal points

Our answer: This has been fixed.

11. Line 227: please rephrase, using the word "but" suggests that a lower median arterial oxygen saturation is associated with lower ESS scores, which it is not. Consider expressing these 2 findings as separate sentences.

Our answer: Two separate sentences have been written. We can now read: “The median arterial oxygen saturation was lower in women and women had a lower ESS score.”

12. Table 4: if changing total registration time to total recording time as discussed above, then correct it in this table, also OAHI to AHI. also correct p values with commas instead of decimal points. Change this in Lines 233 and 234 as well (OAHI to AHI)

Our answer: This has been fixed. 

13. Lines 238 to 239: The numbers do not add up. If 111 had valid data per the Results on line 162 then 6 particants are unaccounted for (40 - No OSA, 65 had OSAHS (30 had mild OSA, 35 with moderate to severe OSA)). Was there any other missing data? I missed any statements on further exclusions after the total of 111.

Our answer: Thank you for the raising this important point. Please, 40 patients without sleep apnea syndrome were compared with 35 patients with moderate to severe OSAHS, and 30 patients have mild OSAHS. Six patients who had central sleep apnea syndrome were excluded from this analysis because our objective was to focus on OSAHS.

14. Lines 244-246. Please explain or rephrase. This sentence is confusing and has no clear meaning as it is written.

Our answer: Thank you for the remark. We presented the results of the logistic regression where we tested some potential factors associated with OSAHS. We did another analysis where we excluded symptoms of OSAHS as potential associated factors, and hypertension is still the only factor significantly associated with OSAHS.

15. Table 5. why age > 58 years? How did you come about 58?? was there something in the data to suggest that 58 was an age that impacted having OSAHS or was it just based on the calculated mean age? I believe age > 50 is what standard questionnaires use. Or if you want to look at teh geriatric population vs the other adults then maybe age of 65 and above etc. 58 appears random. Also, may be better to use "BMI>30" in the last row of the table rather than the term obesity

Our answer: Thank you for the raising this important point. We settled our cut off at 58 years because it is the median age of patients that had been recorded by PMD. We have addressed your remark by modifying the age cut to 50, as seen in the revised table 5. We have also replaced “obesity” by “BMI>30”, as you suggested. 

.

16. Line 287: consider replacing "underline" with "depict"

Our answer: This has been fixed

17. Line 288 - 294: Starting from the sentence "Other studies..."These lines should follow line 284 for the sake of continuity.

Our answer: This has been fixed

18. Lines 285 - 288 ending with ref (14,26). should come after line 294. You may choose to say " Further more in our study....." to bring the readers attention back to the fact that you are referring to the findings from your study.

Our answer: This has been fixed

19. Line 303: Remove "a" before controlled hypertension, did you mean uncontrolled or controlled hypertension?

Our answer: This has been fixed. In that study, OSAHS was assessed in patients with metabolic syndrome and “controlled hypertension”.

20. Line 313: The polysomnogram is not time consuming (takes the same amount of time as a portable study - in both cases overnight). More appropriate to say it is more expensive and requires more resources (skilled staff, sleep lab, equipment etc)

Our answer: This has been fixed and the sentence has been written accordingly.

21. Line 317: Please replaced with AHI. and total recording time

Our answer: This has been fixed.

22. Consider adding to discussion RE: results lines 179 - 180 state that ESS scores did not correlate with sleepiness. This is corroborated by several papers. Examples to look at include: 1. Smith et al . Multiple dimensions of excessive daytime sleepiness J Thorac Dis 2018, 2. Silva et al. identifiction of patients with sleep disordered breathing: comparing the four-variable screening tool... J Clin Sleep med 2011. etc. It may be nice to mention that your study finding corroborates the fact that the ESS should not be used in isolation to determine the presence of sleepiness.

Our answer: Thank you for raising this important point. However in this study we have not objectively assessed sleepiness and we are unable to test the correlation of the ESS score with any objective assessment of sleepiness. 

23. Line 327 - 328 in conclusion: Amend to say "in patients with resistant/uncontrolled hypertension in hospital settings" . It is not cost effective or required in all hospitalized patients with hypertension. Patients with hypertension and risk factors for OSAHS can be studied in the outpatient setting.

Our answer: Thank you very much for the remark that we addressed by modifying the sentence, we can read now: “In hospital setting, the screening for OSAHS in patients with hypertension, particularly if resistant or uncontrolled, is recommended” 

Final comments: Overall a very good paper with excellent potential to publish regarding the OSA experience in Subsaharan Africa.

From the Reviewer 2

1. Have the ESS and STOP-BANG questionnaires been translated and validated in local languages? If so, how does one explain the major discrepancy between the prevalence of sleepiness (assessed qualitatively) and that of ESS>10 (Table 4)?

Our answer: Thank you for this remark. The STOP-BANG and ESS questionnaires were used in well validated French and English versions. No other language was used. It should be emphasized that many studied have found a discrepancy, similar to that in our study, between the prevalence of sleepiness assessed qualitatively and that of ESS>10 [1,2]. It can be explained by the fact that participants may not adequately discriminate fatigue and sleepiness [3].

[1] Vennelle M, Engleman HM, Douglas NJ. Sleepiness and sleep-related accidents in commercial bus drivers. Sleep Breath 2010;14:39–42. doi:10.1007/s11325-009-0277-z.

[2] Al-Abri MA, Al-Adawi S, Al-Abri I, Al-Abri F, Dorvlo A, Wesonga R, et al. Daytime sleepiness among young adult omani car drivers. Sultan Qaboos Univ Med J 2018;18:e143–8. doi:10.18295/squmj.2018.18.02.004.

[3] Neu D, Mairesse O, Hoffmann G, Valsamis J-B, Verbanck P, Linkowski P, et al. Do “sleepy” and “tired” go together? Rasch analysis of the relationships between sleepiness, fatigue and nonrestorative sleep complaints in a nonclinical population sample. Neuroepidemiology 2010;35:1–11. doi:10.1159/000301714.

2. In the study flow diagram - you list 65 invalid PMD records; this is a high rate - one could want to see those repeated, so that power is not affected by the significantly lower sample size analyzed.

Our answer: Thank you for raising this important point. However, as our study involved patients admitted to hospital for a limited period of time, we were unable to repeat the recording in case of invalid initial recording.

3. Abstract (results section) and manuscript: I would avoid causal inferences such as hypertension being the determinant of moderate to severe OSA... It may not be correct and you may not be able to prove it in a cross-sectional study. It is an association.

Our answer: We agree with this comment. We have replaced “determinant” by “associated factor” in the whole manuscript.

4. Abstract (conclusion) and manuscript: too strong a language ('is mandatory...') as a conclusion to this study's findings.

Our answer: We agree too. We have replaced “is mandatory” by “is recommended” in the whole manuscript.

---

## [Editor Report · Decision Letter 1]

20 Dec 2019

PONE-D-19-25768R1

Obstructive sleep apnea and hypopnea syndrome in patients admitted in a tertiary hospital in Cameroon: prevalence and associated factors

PLOS ONE

Dear Dr. Poka-Mayap,

Thank you for submitting your manuscript to PLOS ONE. After careful consideration, we feel that it has merit but does not fully meet PLOS ONE’s publication criteria as it currently stands. Therefore, we invite you to submit a revised version of the manuscript that addresses the points raised during the review process.

ACADEMIC EDITOR: Be sure to:

See below for one final edit. 

We would appreciate receiving your revised manuscript by Feb 03 2020 11:59PM. To enhance the reproducibility of your results, we recommend that if applicable you deposit your laboratory protocols in protocols.io, where a protocol can be assigned its own identifier (DOI) such that it can be cited independently in the future. For instructions see: http://journals.plos.org/plosone/s/submission-guidelines#loc-laboratory-protocols

We look forward to receiving your revised manuscript.

Kind regards,

James Andrew Rowley

Academic Editor

PLOS ONE

Additional Editor Comments (if provided):

As the academic editor, I reviewed the author's revisions based upon the reviewers' comments and find that they have responded in a satisfactory manner.

However, there is one wording change that I have found upon my review that needs changing before full acceptance. Line 336 in the version with the edits shown, the word 'weaker' is not correct. Portable monitoring devices yield a lower than expected AHI given the denominator is recording time and not sleep time but this does not make the AHI 'weaker'. Please word 'Furthermore, PMD yields a lower AHI compared to full polysomnography,' (Note also to take the 'O' out before 'AHI')

---

## [Author Response · Author response to Decision Letter 1]

24 Dec 2019

To the Editors-in-Chief

Plos One 

Submission of a revised manuscript

Manuscript ID: PONE-D-19-25768 entitled " Obstructive sleep apnea and hypopnea syndrome in patients admitted in a tertiary hospital in Cameroon: prevalence and associated factors"

Dear Editor,

We are grateful to the managing editor and reviewers for their time and comments on our manuscript (referenced above). We have addressed their comments and queries and would like to submit an updated version of our manuscript for publication in the Journal. Changes in the main document have been inserted with the use of red and blue color. In addition we are providing below a point-by-point response to each journal requirements. 

Additional Editor Comments 

Please amend your list of authors on the manuscript to ensure that each author is linked to an affiliation.

We note that you have included affiliation numbers 1-7, ¶ and & however only affiliations 1-7 have authors linked to them. Please amend affiliation & and ¶ to link an author to it or remove if added in error.

Our answer: This has been fixed. 

We look forward to the outcome of our submission.

Sincerely,

Virginie Poka-Mayap, Dodo Balkissou Adamou, Massongo Massongo, Steve Voufouo Sonwa, Jacqueline Alime, Ben Patrick Michel Moutlen, Alfred Kongnyu Njamnshi, André Noseda And Eric Walter Pefura-Yone

---

## [Editor Report · Decision Letter 2]

30 Dec 2019

Obstructive sleep apnea and hypopnea syndrome in patients admitted in a tertiary hospital in Cameroon: prevalence and associated factors

PONE-D-19-25768R2

Dear Dr. Poka-Mayap,

We are pleased to inform you that your manuscript has been judged scientifically suitable for publication and will be formally accepted for publication once it complies with all outstanding technical requirements.

With kind regards,

James Andrew Rowley

Academic Editor

PLOS ONE
---

## [Editor Report · Acceptance letter]

8 Jan 2020

PONE-D-19-25768R2 

Obstructive sleep apnea and hypopnea syndrome in patients admitted in a tertiary hospital in Cameroon: prevalence and associated factors 

Dear Dr. Poka-Mayap:

I am pleased to inform you that your manuscript has been deemed suitable for publication in PLOS ONE. Congratulations! Your manuscript is now with our production department. 

With kind regards,

on behalf of

Dr. James Andrew Rowley 

Academic Editor

PLOS ONE